# Corneal In Vivo Laser-Scanning Confocal Microscopy Findings in Dry Eye Patients with Sjögren’s Syndrome

**DOI:** 10.3390/diagnostics10070497

**Published:** 2020-07-20

**Authors:** Yukihiro Matsumoto, Osama M. A. Ibrahim, Takashi Kojima, Murat Dogru, Jun Shimazaki, Kazuo Tsubota

**Affiliations:** 1Department of Ophthalmology, Keio University School of Medicine, Tokyo 160-8582, Japan; ymatsumoto8@aol.com (Y.M.); osamaibrahim82@gmail.com (O.M.A.I.); kojima@chukyogroup.jp (T.K.); tsubota@z3.keio.jp (K.T.); 2Department of Ophthalmology, Ichikawa General Hospital, Tokyo Dental College, Chiba 272-8513, Japan; jun@eyebank.or.jp

**Keywords:** confocal microscopy, cornea, dry eye, Sjögren’s syndrome

## Abstract

Purpose: To evaluate the changes in cornea in Sjögren’s syndrome (SS) with a novel confocal microscopy device. Methods: Twenty-three right eyes of patients with SS (23 women; mean age, 65.4 ± 11.4 years) and 13 right eyes of 13 age- and sex-matched control subjects (13 women; mean age, 68.8 ± 9.8 years) were studied. Furthermore, eight right eyes of patients with SS (8 women; mean age, 66.9 ± 9.6 years) were studied to evaluate the corneal microscopic alterations after the treatment with topical 3% diquafosol sodium eye drops. All cases had tear quantity, tear breakup time (BUT), ocular surface staining measurements, and corneal in vivo laser-scanning confocal microscopy examinations. The density and area of corneal epithelial cells (superficial, wing, and basal), density of corneal stromal cells (anterior, intermediate, and posterior), density and area of corneal endothelial cells, density and morphology of corneal sub-basal nerve plexus, density of corneal sub-basal inflammatory cells were also assessed. Results: The tear quantity, stability, and vital staining scores were significantly worse in patients with SS than in control subjects (*p* < 0.0001). Corneal superficial epithelial cell density was significantly lower in SS compared with control subjects (*p* < 0.0001). Corneal superficial epithelial cell area was significantly larger in SS compared with control subjects (*p* = 0.007). Corneal sub-basal nerve fiber density was lower in SS compared with control subjects (*p* < 0.0001). Morphological abnormality of nerve fibers was observed in SS patients. Corneal sub-basal inflammatory cell density was significantly higher in SS patients compared with control subjects (*p* < 0.0001). Furthermore, the mean corneal superficial epithelial cell density and area, inflammatory cell density, corneal sub-basal nerve fiber density, and morphological abnormality of nerve fibers, were improved with topical 3% diquafosol sodium treatment in the dry eye patients with SS (*p* < 0.05). Conclusions: The diagnostic modality using in vivo laser-scanning confocal microscopy was a useful method for the evaluation of the corneal cell density and area, nerve fiber density and morphology, and inflammatory cell density in patients with SS and also a useful tool in the assessment of treatment effect with topical 3% diquafosol sodium in the SS patients.

## 1. Introduction

The pathogenesis of Sjögren’s syndrome (SS) still remains a mystery today while genetic and environmental factors have been implicated as important [1]. SS mostly involves females between 40–50 years of age, and the incidence has been reported to be as high as 1/100 and as low as 1/1000 [2,3].

It is well-known that SS causes a tear-deficient dry eye, which leads to the ocular surface epithelial damage. Inflammation is a central mechanism in SS leading to decreased lacrimal secretion and increased keratoconjunctival vital staining [4,5,6]. The SS patients with ocular surface damage have various ocular symptoms such as discomfort, which have a bad effect on the quality of life. In accordance with the other studies in the literature [7,8,9,10], we previously reported significantly lower tear quantity and stability values and ocular surface epithelial damage scores in patients with SS when compared with control subjects [1]. However, it has been difficult to study corneal microscopic alterations including the epithelial damage at the cellular level since methods such as impression cytology can be invasive [11].

The international Dry Eye Workshop II (DEWS II) reported in 2017 that “dry eye is a multifactorial disease of the ocular surface characterized by a loss of homeostasis of the tear film, and accompanied by ocular symptoms, in which tear film instability and hyperosmolarity, ocular surface inflammation and damage, and neurosensory abnormalities play etiological roles” [12]. The assessment of the histopathology to evaluate the ocular surface inflammation, damage, and neurosensory abnormalities that play etiological roles in dry eye has been problematic.

Confocal microscopy is a novel non-invasive technique that allows investigation of ocular surface—lacrimal gland unit at the cellular level [13,14,15,16,17,18,19,20,21]. Especially, the application is helpful to diagnose in real time the infectious keratitis cases such as Acanthamoeba and fungi [15,16]. Technological advances allowed better quality images in various ocular surface disorders. The ability of in vivo confocal microscopy has been employed in investigation of the presence of inflammation and cytological changes in conjunction with ocular surface examinations such as vital staining, impression, and brush cytology [7,8,9,10,13,14,15,16,17,18,19,20,21]. In vivo laser-scanning confocal microscopic observations of the conjunctiva in patients with SS revealed a decreased conjunctival epithelial cell density, increased conjunctival epithelial microcyst density, and increased conjunctival inflammatory cell density in a previous study [1].

In this study, we used in vivo laser-scanning confocal microscopy to evaluate the morphologic changes of corneal cells, nerves, and inflammatory cells in patients with SS and compared the results with those in healthy control subjects. We also investigated the corneal alterations in SS patients after the treatment with topical 3% diquafosol sodium as assessed by confocal microscopy.

## 2. Materials and Methods

### 2.1. Subjects and Examinations

Twenty-three right eyes of patients with SS (23 women; mean age, 65.4 ± 11.4 years) and 13 right eyes of 13 age- and sex-matched control subjects (13 women; mean age, 68.8 ± 9.8 years) were studied to evaluate the corneal microscopic alterations in patients with SS. Furthermore, eight right eyes of patients with SS (8 women; mean age, 66.9 ± 9.6 years) were studied to evaluate the corneal microscopic alterations in patients with SS after the treatment with topical 3% diquafosol sodium eye drops (Diquas^®^, Santen Pharmaceutical Co. Ltd., Osaka, Japan), applied 6 times per day for 3 months.

The SS diagnosis was made according to the Japanese consensus criteria. The diagnosis of SS in Japan is based on the presence of any two of the following four criteria: (1) at least one of the following ocular signs: Schirmer’s test result less than 5 mm/5 min and Rose Bengal staining score > 3 points (in the van Bijsterfeld scoring) or fluorescein positive staining; (2) at least one of the following oral signs: sialography > stage 1 (in the Rubin and Holt grading), or salivary gland hyposecretion (gum test < 10 mL/10 min, or Saxon test < 2 g/2 min) and salivary gland dysfunction (salivary gland scintigraphy); (3) pathological evidence of inflammation in the lacrimal or minor salivary gland > grade 4 (in the Greenspan grading); (4) serological evidence of autoantibodies in the serum to SS-A(Ro) or SS-B(La) antigens or both [22]. The SS patients were also diagnosed with aqueous tear deficient dry eye.

The diagnosis of dry eye was made by the following the Japanese consensus criteria: symptoms of dry eye, abnormality of tear production as determined by the Schirmer test (<5 mm/5 min), tear film instability as determined by the break-up time (BUT) (<5 s), and positive ocular surface vital staining (with fluorescein and Rose Bengal dye) [23]. None of the subjects had a history of ocular surgery, other ocular or systemic disease, or contact lens use that would alter the ocular surface. Informed consent was obtained from all subjects. Examination procedures were board reviewed (Tokyo Dental College, I 15-88, 30 May 2016), and the study was conducted in accordance with the tenets of the Declaration of Helsinki.

### 2.2. Tear Function Test and Ocular Surface Vital Staining

The tear film BUT measurement and ocular surface vital staining test were performed with slit-lamp microscope. Rose Bengal (1%) and fluorescein dye (1%) were used for ocular surface vital staining. Tear film BUT measurement, vital staining scoring of the ocular surface, and Schirmer test have been performed according to a previously published protocol [1,24].

### 2.3. In Vivo Laser-Scanning Confocal Microscopy

All cases underwent in vivo laser-scanning confocal microscopy with the Heidelberg Retina Tomograph II, Rostock Corneal Module [HRT II-RCM] (Heidelberg Engineering GmbH, Dossenheim, Germany). The details of this examination have been reported previously [1].

### 2.4. Corneal Image Analysis

At least five sequences (100 images per sequence) of corneal images were taken for each eye. The morphologic characteristics of the cornea were observed in patients with SS and compared with those of normal control subjects. The cornea in the epithelial layer was divided at three different depths (superficial epithelial cells, <10 μm; intermediate epithelial cells, 10–40 μm; basal epithelial cells, >40 μm) and also in the stromal layer at three different depths (anterior stromal cells, <100 μm; intermediate stromal cells, 100–400 μm; posterior stromal cells, >400 μm) in this study. We chose the sub-basal epithelial layer to analyze the inflammatory cell densities, because the inflammatory cells appeared like dendritic hyper-reflective bodies with the sub-basal epithelial area providing an excellent contrast for inflammatory cell counts, and also chose the sub-basal epithelial layer to analyze the density and morphology of sub-basal nerve plexus. Each cell was manually marked with a blue dot inside each 400 × 400 μm frame, and the cell counts were performed automatically, and the cell density was calculated in cells/mm^2^ (Cell Count software; Heidelberg Engineering GmbH). The length of nerve fiber in μm and cell area in μm^2^ were calculated using the Image J software (Java software program developed by the National Institutes of Health, (Bethesda, MD, USA).

The sub-basal epithelial nerve plexus was observed in the layer just anterior to Bowman’s layer. The morphological study of the corneal nerves was analyzed by the following parameters; (1) Number of nerves (nerves/frame): defined as the sum of the nerve branches present in one image. (2) Density of nerves (μm/mm^2^): defined as the total length of the nerve fibers per square millimeter. (3) Number of beadings (beadings/100 μm of nerve fiber): defined as the number of bead-like formations in 100 μm of nerve fiber. (4) Nerve tortuosity (grade): classified in four grades according to the previous reported scale [25]. (5) Nerve reflectivity (grade): classified in four grades according to the previous reported scale [25].

At least three randomized, non-overlapping, digital images with high resolution and contrast from a single sequence were selected for analysis from the all target layers of each eye. The images were selected by a co-researcher who was masked to whether the subject was a patient with SS or a control subject. The results were expressed as the mean ± standard deviation.

### 2.5. Statistical Analysis

The Student t-test was applied to test the statistical differences in tear film BUT, Schirmer test value, cell density, cell area, and age, between the SS patients and control subjects. The Mann–Whitney test was used to study the differences in fluorescein staining score, Rose Bengal staining score, and corneal nerve morphological grade, between the SS patients and control subjects (Instat software, version 3.0; GraphPad, San Diego, CA, USA). A probability of less than 5% was considered statistically significant.

## 3. Results

Statistical analyses did not reveal differences in relation to age and gender between controls and patients (*p* = 0.27).

### 3.1. Tear Functions and Ocular Surface Staining Scores

Comparisons of tear functions and ocular surface findings between the normal control subjects and the patients with SS are shown in Table 1. The mean Schirmer test value was significantly lower in the patients with SS than in the normal control subjects (*p* < 0.0001). The mean BUT was significantly lower in the patients with SS than in the normal control subjects (*p* < 0.0001). The mean fluorescein and Rose Bengal staining scores were significantly higher in the patients with SS when compared with the normal control subjects (*p* < 0.0001, *p* < 0.0001, respectively).

Changes of tear functions and ocular surface findings before and after the treatment are shown in Table 2. The mean BUT, fluorescein, and Rose Bengal staining scores, tended to improve with topical 3% diquafosol sodium eye drops but did not show significant differences before and after treatment (*p* = 0.13, *p* = 0.14, *p* = 0.27, respectively).

### 3.2. In Vivo Laser-Scanning Confocal Microscopy

#### 3.2.1. Corneal Cell Density and Area

Comparisons of confocal microscopic parameters between the normal control subjects and the patients with SS are shown in Table 1. The mean corneal superficial epithelial cell density was significantly lower in the patients with SS, compared with the healthy control subjects (*p* < 0.0001). The mean corneal superficial epithelial cell area was significantly larger in the patients with SS, compared with the healthy control subjects (*p* = 0.002). There were no significant differences in the mean corneal intermediate (wing) epithelial cell densities between the patients with SS and the healthy control subjects (*p* = 0.62) and in the mean corneal intermediate (wing) epithelial cell areas between the patients with SS, compared with the healthy control subjects (*p* = 0.10). There were also no significant differences in the mean corneal basal epithelial cell densities between the patients with SS and the healthy control subjects (*p* = 0.20) and in the mean corneal basal epithelial cell areas between the patients with SS and the healthy control subjects (*p* = 0.28).

There was a significant difference in the mean corneal anterior stromal cell (keratocyte) density between the patients with SS and the healthy control subjects (*p* = 0.04); however, there were no significant differences in the mean corneal intermediate stromal cell (keratocyte) densities between the patients with SS and the healthy control subjects (*p* = 0.06) and in the mean corneal posterior stromal cell (keratocyte) densities between the patients with SS and the healthy control subjects (*p* = 0.77).

In addition, no significant differences in the mean corneal endothelial cell densities between the patients with SS and the healthy control subjects (*p* = 0.27) and in the mean corneal endothelial cell areas between the patients with SS and the healthy control subjects (*p* = 0.17) were noted.

Changes of confocal microscopic parameters before and after the treatment are shown in Table 2. The mean corneal superficial epithelial cell density significantly increased after the treatment with topical 3% diquafosol sodium eye drops, compared to before the treatment in the patients with SS (*p* = 0.001). The mean corneal superficial epithelial cell area also significantly decreased after the treatment, compared to before the treatment (*p* = 0.01). There were no significant differences in the mean corneal intermediate (wing) epithelial cell densities and areas before and after the treatment (respectively; *p* = 0.10, *p* = 0.28). There were also no significant differences in the mean corneal basal epithelial cell densities and areas before and after the treatment (*p* = 0.87, *p* = 0.94, respectively). There were also no significant differences in the mean corneal anterior, intermediate, and posterior stromal cell (keratocyte) densities before and after the treatment (respectively; *p* = 0.94, *p* = 0.96, *p* = 0.54). In addition, no significant differences in the mean corneal endothelial cell densities and areas before and after the treatment (*p* = 0.63, *p* = 0.36, respectively) were observed.

#### 3.2.2. Corneal Inflammatory Cell Density

The mean inflammatory cell density at the sub-basal epithelial area was significantly higher in the patients with SS, compared with the healthy control subjects (*p* < 0.0001) as shown in Table 1. The mean inflammatory cell density at the sub-basal epithelial area significantly decreased after the treatment, compared to the density before the treatment (*p* = 0.03) as shown in Table 2.

#### 3.2.3. Corneal Nerve Density and Morphology

The mean number of corneal nerves was significantly lower in the patients with SS than in the healthy control subjects (*p* < 0.0001). The mean corneal nerve density was also significantly lower in the patients with SS than in the healthy control subjects (*p* = 0.0001). The mean number of beadings was significantly higher in the patients with SS, than in the healthy control subjects (*p* < 0.0002). The nerve tortuosity and reflectivity values were higher in the patients with SS, compared with the healthy control subjects, with a significant difference (*p* < 0.0001, *p* = 0.04, respectively).

The mean number of corneal nerves significantly increased after the treatment, compared to the mean number before the treatment (*p* = 0.04). The mean corneal nerve density also significantly increased after the treatment, compared to the density before the treatment (*p* = 0.02). The mean number of beadings significantly decreased after the treatment (*p* = 0.0008). A significant decrease in nerve tortuosity was observed after treatment (*p* = 0.04) but no significant change in nerve reflectivity (*p* = 0.18).

## 4. Discussion

SS is an exocrine disease in which the lacrimal and salivary glands are targeted mainly by an autoimmune process. The dry eye accompanied by SS is one of the most serious aqueous-deficient dry eye diseases and induces corneal and conjunctival epithelial damage.

In this study, we evaluated the morphological changes in the corneal epithelial, stromal, and endothelial cells, sub-basal nerve fibers, and inflammatory cells in patients with SS, comparing the results with healthy controls. We also investigated the changes in the aforementioned parameters with topical 3% diquafosol sodium treatment.

In previous reports using an in vivo white-light slit-scanning confocal microscope by Benitez-del-Castillo JM et al. and Villani E et al. [7,8,9,10], the corneal superficial epithelial cell density was shown to be significantly lower in SS patients, compared with the healthy control subjects. Villani E et al. reported significantly higher corneal basal epithelial cell density in the SS patients [9,10]; however, Benitez-del-Castillo JM et al. reported no significant difference between the SS patients and the healthy control subjects [7,8]. In our study using an in vivo laser-scanning confocal microscope, the mean corneal epithelial cell density including superficial, wing, and basal cells, tended to be lower and the mean cell area tended to be larger in the SS patients, compared with the healthy control subjects. There was a significant change of both cell density and area in the superficial epithelial cells; however, no significant change in the wing and basal epithelial cells. As shown in Figure 1, loss of superficial epithelial cells in the dark areas nearby enlarged cells was observed in the SS patients. The superficial epithelial cell loss was thought to be caused by apoptotic changes in the serious aqueous-deficient dry eye diseases and can result in positive-fluorescein staining with the vital corneal staining tests.

An increase of corneal superficial epithelial cell size has been reported using a specular microscope in the healthy subjects with extended wear soft contact lenses [26,27]. The mechanism of enlargement of superficial epithelial cells after extended wear soft contact lenses use is not clear; however, possibilities include altered mitotic rate caused by the lower oxygen pressure, accelerated differentiation, and sustained desquamation induced by mechanical coverage. The mechanism of enlargement and loss of superficial epithelial cells in the SS patients is also not yet fully understood. Dry eye disease can induce the up-regulation and secretion of pro-inflammatory mediators including interleukin (IL)-1α, IL-1β, IL-6, IL-8, transforming growth factor (TGF)-β1, and tumor necrosis factor (TNF)-α, which are produced in the epithelial cells and lymphocytes and secreted into the tear film [4,5,6]. Elevation of various cytokine levels within the tear film, combined with the inflammation in the lacrimal gland, and reduced concentration of essential lacrimal gland-derived factors such as epidermal growth factor (EGF), can create an environment in which terminal differentiation of the ocular surface epithelium is impaired [6]. Moreover, the integrity of ocular surface epithelial cells depends on the presence of intact innervations; however, the diminishment of corneal sub-basal nerves is observed in dry eye disease [7,8,9,10]. The reduced concentration of neurotrophic factor such as nerve growth factor (NGF), which is synthesized and secreted at the terminal end of nerve fibers, can down-regulate the corneal epithelial proliferation and integrity.

In previous reports by Benitez-del-Castillo JM et al. and Villani E et al., a significant increase of corneal anterior keratocyte density in SS patients, compared with the healthy control subjects, was shown [7,8,9,10]. Villani E et al. reported a significant increase of corneal posterior keratocyte density in the SS patients [9,10]; however, Benitez-del-Castillo JM et al. reported no significant changes between the SS patients and the healthy control subjects [7,8]. The hypothesis is that chronic inflammation induced by pro-inflammatory cytokines such as IL-1, IL-6, leads to activation of keratocytes, which synthesize NGF or other factors of nerve growth in the patients with SS [7,8,9,10]. The activated keratocytes, which showed hyperreflective stromal cell images, are able to produce and secrete NGF, thus making a relevant contribution to the processes of activation and reorganization of the sub-basal nerve plexus in the patients with SS. Moreover, there is a hypothesis of corneal stromal thinning due to inflammatory processes. Indeed, hyper-production of TNF-α and IL-1 would be responsible for apoptosis as well as for an increase in the proteolytic activity at the stromal level in patients with SS [28]. However, in our study, the keratocyte densities in the anterior, intermediate, and posterior stroma, tended to be lower in the SS patients, compared with the healthy control subjects. Especially, there was a significant change of anterior keratocyte density. It is well-known that the keratocytes activated by inflammation may transdifferentiate into the myofibroblast cells, which can produce the collagen fiber and result in fibrosis of cornea [29,30]. NGF can modulate some functional activities of fibroblastic-keratocytes [31], and the NGF eyedrops induce the healing of neurotrophic or autoimmune corneal ulcers [32,33]. We hypothesize that the diminishment of corneal sub-basal nerves may influence the decrease of keratocyte density in the patients with SS, because the keratocytes as well as the epithelial cells are innervated. The decrease of neurotrophic factors such as NGF can down-regulate the corneal stromal integrity. The keratocyte loss tends to be remarkable at the anterior stroma near the sub-basal nerve plexus in the patients with SS. Moreover, pro-inflammatory mediators such as matrix metalloproteinase (MMP)-9 may induce corneal stromal thinning through increased proteolytic activity at the stromal level in dry eyes [34].

To our knowledge, no research has been reported in regard to the corneal endothelial cell density and area, which were measured by confocal microscopy in patients with SS. Our study showed no significant changes in both corneal endothelial cell densities and areas between the SS patients and the healthy control subjects.

The density of the dendritic cells in the cornea of patients with SS was reported to be significantly higher than that of healthy subjects by Wakamatsu TH et al. and Lin H et al. [1,35]. This study revealed a similar outcome with the previous reports. The number of inflammatory dendritic cells at the sub-basal epithelial area dramatically increased in the SS patients, approximately by five times compared to the healthy control subjects. The dendritic antigen-presenting cells play a critical role in corneal immunology and have been observed by in vivo confocal microscopy in both healthy and affected corneas. The migration and maturation of dendritic cells are activated in response to pro-inflammatory stimulation [36,37]. We believe that the dendritic cells are associated with the ocular surface inflammation in dry eye accompanied with SS.

Benitez-del-Castillo JM et al. and Villani E et al. reported a significant decrease of the sub-basal nerve number in the SS patients [7,8,9,10]; however, both Tuominen ISJ et al. and Tuisku IS et al. reported no significant differences between the SS patients and control subjects [38,39]. Benitez-del-Castillo JM et al. also reported a significant decrease of the sub-basal nerve density in the SS patients [7,8]. In this study, the corneal sub-basal epithelial nerve number and density were observed to be significantly decreased in the SS patients, compared with the healthy control subjects. The corneal nerves are mainly derived from the ophthalmic branch of the trigeminal nerve and terminate as free nerve endings between the epithelial cells. Even single epithelial and stromal cells may be innervated. The nerve fibers have an important influence in the corneal trophism and contribute to the maintenance of healthy cornea. It was reported that there was a strong and significant correlation between the increased dendritic cells and the decreased sub-basal corneal nerves in infectious keratitis, suggesting a potential interaction between the immune and nervous systems in the cornea [40]. It was also reported that the dendritic (immune) cells mediated clearance of axonal debris from intraepithelial corneal basal nerves after injury [41]. In other words, while the dendritic (immune) cells may secrete the cytokines that induce the neural axon loss, they also may play roles in removal of axon debris. This is another explanation for the positive correlation between the increased dendritic cells and decreased intraepithelial corneal basal nerves. In addition, Stepp MA et al. recommended the changes from the canonical “sub-basal nerves” to the proposed amended “intraepithelial corneal basal nerves” regarding the corneal nerve terminology in morphology [42]. We postulate that the inflammatory cells may induce the diminishment of nerve fibers in the sub-basal nerve plexus in dry eye patients with SS. As shown in Figure 2, many inflammatory cells were observed near the sub-basal nerve fibers in the SS patients. Xu KP et al. showed that the corneal sensitivity decreased in dry eyes with SS [43]. Benitez-del-Castillo JM et al. and Villani E et al. also reported that there was a statistically significant correlation between the corneal sensitivity and the corneal sub-basal nerve numbers or densities [8,10]. In our opinion, the diminishment of the corneal nerves may reduce the secretion of neurotrophic factors and lead to the apoptotic cell death in the epithelium and stroma. The ocular surface disorders caused by the alteration of corneal innervation in dry eyes with SS appear to be a type of neurotrophic keratopathy.

Benitez-del-Castillo JM et al. and Villani E et al. reported a significant increase in the number of beadings and nerve tortuosity, but no significant changes in the nerve reflectivity in the SS patients [7,9,10]. The morphological changes such as beadlike formations, tortuosity, and reflectivity are considered to be indices of metabolic activity of the nerve plexus. The general belief is that the overexpression of NGF, which can be produced by the activated keratocytes, may contribute to the process of activation, reorganization, and hypertrophy of the peripheral nerve fibers [7,8,9,10,38]. Our study showed a significant increase in the number of beadings, nerve tortuosity, and nerve reflectivity in the SS patients. Figure 3 showed that very tortuous sub-basal nerve fibers were observed in the SS patients. Pro-inflammatory mediators such as TNF-α and IL-1, which are produced in the astrocytes, have been shown to induce neuro-inflammation and result in nerve damage in the brain of the patients with cerebral infarction [44,45]. We hypothesize that the pro-inflammatory mediators such as TNF-α and IL-1, which are secreted from sub-basal inflammatory cells, may lead to the morphological change in sub-basal corneal nerve plexus of the patients with SS. Figure 2 showed that many inflammatory cells were observed near the sub-basal nerve fibers in the SS patients. Chang PY et al. also observed morphological changes in the sub-basal nerve plexus with decreased corneal epithelial cell and nerve fiber density in patients with diabetes, who possibly had neurotrophic corneal diseases [46]. We postulate that the morphological changes may be caused by not only chronic inflammation but also corneal denervation.

Topical 3% diquafosol sodium is a commercially available ophthalmic solution for the treatment of dry eye. Diquafosol has been reported to be an agonist of the purinergic P2Y_2_ receptor that is expressed in several ocular tissues including conjunctival epithelium and goblet cells [47,48,49]. At the cellular level, the P2Y_2_ receptor is known to contribute to water transfer and mucin secretion. Topical 3% diquafosol sodium has been reported to be effective and safe in the treatment of aqueous tear deficient type dry eye patients with SS [50,51].

The impact of various treatment modalities in dry eyes can be effectively assessed by in vivo confocal microscopes [52]. While topical corticosteroids eye drops used for a month were not reported to influence the morphology of corneal sub-basal nerve fibers by Kheirkhah A et al. [53], Villani E et al. found a decline in dendritic cell and activated keratocyte numbers with topical corticosteroids in dry eye patients using HRT-RCM [54]. A report by Iaccheri B et al. showed marked changes in corneal epithelial and nerve fiber morphology (an increase in corneal intermediate epithelial cell number, a decrease in activated keratocyte number, and a decrease in density, tortuosity, and reflectivity of corneal nerve fibers) with topical 0.05% cyclosporine eye drops for 6 months in dry eye patients [55]. Levy O et al. using the same eye drops showed significantly an increase in corneal sub-basal nerve fibers and a decrease in dendritic cell numbers within 6 months after initiation of treatment in SS patients using HRT-RCM [56].

In our study, in vivo confocal microscopic parameters, including the mean corneal superficial epithelial cell density and area, inflammatory cell density, number of nerves, density of nerves, number of beadings, and nerve tortuosity, were improved after treatment with topical 3% diquafosol sodium in dry eye patients with SS. Our study revealed interesting corneal morphologic changes after treatments with topical 3% diquafosol sodium in dry eye patients with SS using HRT-RCM. In the future, in vivo confocal microscopic parameters, in conjunction with Schirmer’s test values and ocular surface vital staining scores, may constitute to diagnostic criteria of dry eyes if the confocal microscopic cut-off values are established, which may also help in the assessment of treatment effectiveness for dry eye disease. Moreover, in vivo confocal microscopy can be used to evaluate the efficacy of other new treatment options such as topical 2% rebamipide for dry eye disease.

In summary, we evaluated the morphologic changes of the corneal cells and nerves in dry eye patients with SS using in vivo laser-scanning confocal microscopy. We found several alterations including the corneal epithelial and stromal cell loss, sub-basal nerve fiber denervation, and abnormal morphology, with increased inflammatory cell populations. The corneal microscopic alterations are considered to be caused by the ocular surface dryness in patients with SS. Indeed, the ocular surface status in dry eye is affected by the pro-inflammatory mediators such as IL-1, IL-6, TGF-β, TNF-α, and MMP-9; neurotrophic factors such as NGF; and antigen-presenting cells. We postulate that there are important relations between the corneal epithelial cell loss and pro-inflammatory mediators, between the epithelial or stromal cell loss and neurotrophic factors, and between the nerve fiber denervation or abnormal morphology and inflammatory cell populations. In the future, we need to confirm these relations by means of other clinical studies using in vivo confocal microscopy.

## 5. Conclusions

In conclusion, this diagnostic modality using in vivo laser-scanning confocal microscopy was a useful method for the evaluation of the corneal cell density and area, nerve fiber density and morphology, and inflammatory cell density in patients with SS. It was also useful tool to evaluate the safety and efficacy of treatment in the dry eye patients with SS.

## Figures and Tables

**Figure 1 diagnostics-10-00497-f001:**
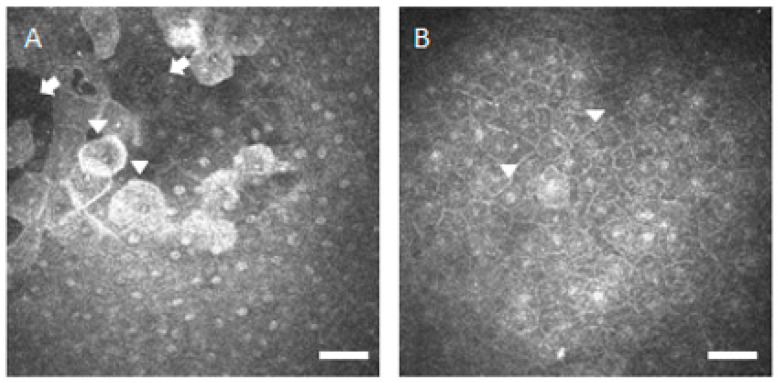
Representative in vivo confocal microscopic corneal superficial epithelial images. (**A**) Image from patient with Sjögren’s syndrome shows enlarged superficial epithelial cells (arrowheads) and loss of superficial epithelial cells in the dark areas (arrows). Bar = 50 μm. (**B**) Image from healthy control subject shows small and compact superficial epithelial cells (arrowheads). Bar = 50 μm.

**Figure 2 diagnostics-10-00497-f002:**
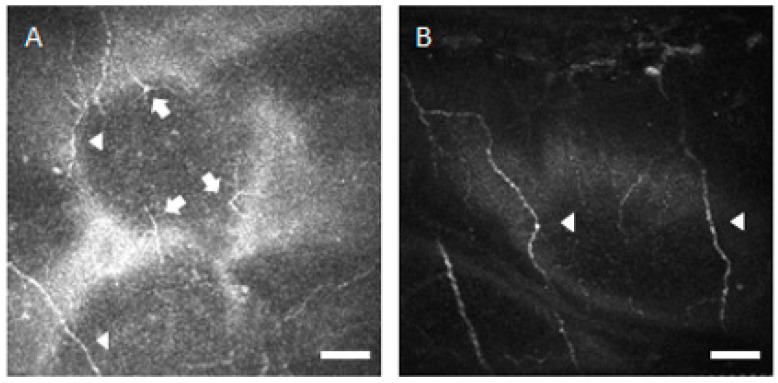
Representative in vivo confocal microscopic corneal sub-basal inflammatory cell images. (**A**) Image from patient with Sjögren’s syndrome shows a sub-basal nerve fiber (arrowhead) and many dendritic inflammatory cells (arrows). Bar = 50 μm. (**B**) Image from healthy control subject shows a sub-basal nerve fiber (arrowhead) and an absence of inflammatory cells. Bar = 50 μm.

**Figure 3 diagnostics-10-00497-f003:**
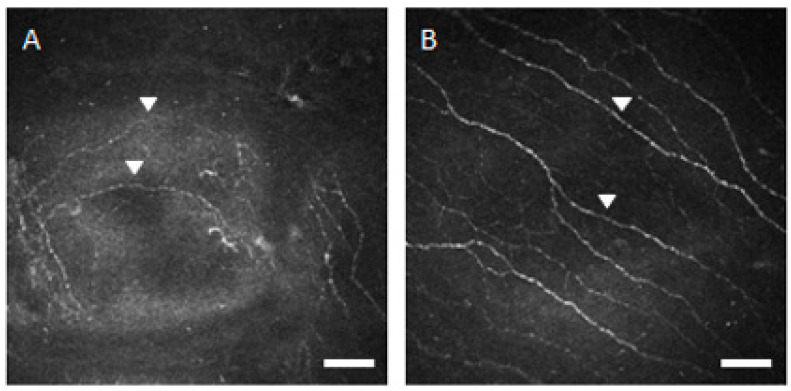
Representative in vivo confocal microscopic corneal sub-basal nerve images. (**A**) Image from patient with Sjögren’s syndrome shows very tortuous sub-basal nerve fibers classified in grade 4 (arrowheads). Bar = 50 μm. (**B**) Image from healthy control subject shows slightly tortuous sub-basal nerve fibers classified in grade 1 (arrowheads). Bar = 50 μm.

**Table 1 diagnostics-10-00497-t001:** Comparisons of tear functions, ocular surface vital staining scores, and in vivo laser-scanning confocal microscopic data.

	Sjögren’s Syndrome	Healthy Control	Probability
Schirmer test (mm)	2.3 ± 1.4	9.0 ± 4.7	<0.0001
Fluorescein staining score (points)	3.8 ± 1.8	0.2 ± 0.6	<0.0001
Rose Bengal staining score (points)	5.0 ± 1.9	0.1 ± 0.3	<0.0001
Break-up time (seconds)	3.3 ± 1.0	11.1 ± 6.1	<0.0001
Superficial epithelial cell density (cells/mm^2^)	829.7 ± 188.5	1215.9 ± 265.8	<0.0001
Superficial epithelial cell area (μm^2^)	898.5 ± 167.0	717.9 ± 150.5	=0.002
Intermediate epithelial cell density (cells/mm^2^)	3770.0 ± 521.1	3850.5 ± 360.7	=0.62
Intermediate epithelial cell area (μm^2^)	193.7 ± 30.1	168.4 ± 29.1	=0.10
Basal epithelial cell density (cells/mm^2^)	6087.9 ± 399.8	6301.1 ± 596.2	=0.20
Basal epithelial cell area (μm^2^)	116.8 ± 22.5	109.8 ± 19.6	=0.28
Anterior stromal cell density (cells/mm^2^)	477.0 ± 47.7	529.1 ± 93.9	=0.04
Intermediate stromal cell density (cells/mm^2^)	393.3 ± 45.1	429.8 ± 58.7	=0.06
Posterior stromal cell density (cells/mm^2^)	333.6 ± 33.6	338.0 ± 53.3	=0.77
Endothelial cell density (cells/mm^2^)	3126.3 ± 237.0	3294.2 ± 330.0	=0.27
Endothelial cell area (μm^2^)	319.1 ± 25.1	289.4 ± 47.9	=0.17
Sub-basal inflammatory cell density (cells/mm^2^)	87.0 ± 52.5	17.3 ± 18.8	<0.0001
Sub-basal nerve number (nerves/frame)	3.57 ± 0.62	5.12 ± 0.85	<0.0001
Sub-basal nerve density (μm/mm^2^)	1145.6 ± 385.0	1799.9 ± 427.4	<0.0001
Sub-basal nerve beadings number (beadings/100 μm)	10.2 ± 2.5	6.6 ± 2.3	<0.0002
Sub-basal nerve tortuosity (grade)	2.91 ± 0.75	1.63 ± 0.57	<0.0001
Sub-basal nerve reflectively (grade)	2.13 ± 0.55	1.73 ± 0.41	=0.04

**Table 2 diagnostics-10-00497-t002:** Changes of tear functions, ocular surface vital staining scores, and in vivo laser-scanning confocal microscopic data in patients with Sjögren’s syndrome by the treatment #1.

	Before Treatment	After Treatment	Probability
Fluorescein staining score (points)	3.1 ± 1.6	2.3 ± 2.1	=0.13
Rose Bengal staining score (points)	4.9 ± 3.7	3.7 ± 1.9	=0.14
Break-up time (seconds)	2.1 ± 1.1	3.6 ± 1.5	=0.27
Superficial epithelial cell density (cells/mm^2^)	1001.1 ± 170.6	1396.4 ± 207.7	=0.001
Superficial epithelial cell area (μm^2^)	790.0 ± 121.4	599.1 ± 76.4	=0.01
Intermediate epithelial cell density (cells/mm^2^)	4043.8 ± 410.6	4503.4 ± 392.5	=0.10
Intermediate epithelial cell area (μm^2^)	152.5 ± 21.8	138.9 ± 18.3	=0.28
Basal epithelial cell density (cells/mm^2^)	6697.9 ± 483.7	6628.9 ± 655.7	=0.87
Basal epithelial cell area (μm^2^)	94.6 ± 16.3	91.1 ± 16.2	=0.94
Anterior stromal cell density (cells/mm^2^)	446.1 ± 60.3	456.8 ± 86.8	=0.94
Intermediate stromal cell density (cells/mm^2^)	352.6 ± 34.4	349.2 ± 13.0	=0.96
Posterior stromal cell density (cells/mm^2^)	306.9 ± 22.5	323.2 ± 21.9	=0.54
Endothelial cell density (cells/mm^2^)	3245.1 ± 243.2	3434.0 ± 331.4	=0.63
Endothelial cell area (μm^2^)	289.1 ± 23.4	267.2 ± 15.8	=0.36
Sub-basal inflammatory cell density (cells/mm^2^)	117.3 ± 53.7	68.0 ± 35.2	=0.03
Sub-basal nerve number (nerves/frame)	2.63 ± 0.96	3.96 ± 1.07	=0.04
Sub-basal nerve density (μm/mm^2^)	829.6 ± 348.0	1238.6 ± 410.1	=0.02
Sub-basal nerve beadings number (beadings/100 μm)	10.7 ± 2.3	7.9 ± 1.8	=0.0008
Sub-basal nerve tortuosity (grade)	1.78 ± 0.38	1.19 ± 0.38	=0.04
Sub-basal nerve reflectively (grade)	2.14 ± 0.30	1.89 ± 0.38	=0.18

#1 Topical 3% diquafosol sodium (Diquas^®^), applied 6 times per day for 3 months.

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
