# Peer review of "Corneal In Vivo Laser-Scanning Confocal Microscopy Findings in Dry Eye Patients with Sjögren’s Syndrome"

_diagnostics, 2020, doi:10.3390/diagnostics10070497_

Round 1
Reviewer 1 Report
The study by Matsumoto and colleagues titled “Corneal in vivo Laser-Scanning Confocal Microscopy Findings in Dry Eye Patients with Sjogren’s Syndrome” is well written and will be of interest to those treating patients with dry eye disease. There are issues that may help the study be appreciated by our colleagues.
- The introduction is quite short (less than ½ page) and the discussion is just over 4 pages. The authors should consider moving some of the text and issues raised in the discussion to the introduction and leave the discussion shorter to focus their major points.
- Tables 1 and 3 show various values for assessments made for control and SS patients; Tables 2 and 4 show values for assessments made for SS patients “before treatment” and after treatment with 3% diquafosal sodium for 3 months. The SS values listed for patients on Table 1 and 2 are different from those listed for SS patients before treatment on Tables 2 and 4. The authors should clarify this issue. The differences likely reflect different populations of patients.
- This reader kept referring back to controls while reading Tables 2 and 4. If possible, authors should consider combining Table 1 and 2 and Tables 2 and 4 to allow readers to see that while values for FL, RB, and BUT in Tables 1 and 2 and for inflammatory cells and sub-basal axon density for SS patients improve with treatment, most of these values remain significantly different from controls. One exception is superficial epithelial cell density and area which appear to be similar for SS patients after treatment compared to controls.
- Another explanation for the positive correlation between dendritic cells and axon loss is the role that dendritic cells play in clearing axonal debris. As debris accumulates, dendritic cells activate to assist the corneal epithelial cells in removing it (see paper titled "Axonal Debris Accumulates in Corneal Epithelial Cells After Intraepithelial Corneal Nerves Are Damaged: A Focused Ion Beam Scanning Electron Microscopy (FIB-SEM) Study' in Exp Eye Res 2020 May; 194:107998). While immune cells may secrete cytokines that induce axon loss, they also may play roles in clearing debris after axon loss.
- The authors use the classic canonical term for the sub-basal nerves. Consideration should be given to using the amended terminology proposed in the paper titled Corneal Epithelial “Neuromas”: A Case of Mistaken Identity? In the journal Cornea 2020;39:930–934. The "sub-basal nerves" and "sub-basal inflammatory cells" are actually located within the epithelium at the basal aspect of the corneal epithelial basal cells above the epithelial basement membrane and Bowman's layer. Using the classical terms can make it more difficult for colleagues to appreciate the anatomical relationships between the corneal epithelial cells, dendritic cells, basement membrane and Bowman's layer.
Author Response
Response to reviewer 1
Thank you for your time in reviewing our manuscript. We tried to revise the paper in accordance with the criticisms of the reviewer within our best providing point-by-point responses as follows:
Comments and Suggestions for Authors
The study by Matsumoto and colleagues titled “Corneal in vivo Laser-Scanning Confocal Microscopy Findings in Dry Eye Patients with Sjogren’s Syndrome” is well written and will be of interest to those treating patients with dry eye disease. There are issues that may help the study be appreciated by our colleagues.
- The introduction is quite short (less than ½ page) and the discussion is just over 4 pages. The authors should consider moving some of the text and issues raised in the discussion to the introduction and leave the discussion shorter to focus their major points.
(Reply 1)
Thank you so much for your constructive comments. According to the reviewer’s requests, we moved the below sentences (No.1,2,4,5) from the discussion section to the introduction section. Furthermore, we added the below sentence (No.3) in the introduction section. Please note that the discussion has now been shortened and the introduction is longer as requested.
1.“ In accordance with the other studies in the literature [1,7-10], we previously reported significantly lower tear quantity and stability values, and ocular surface epithelial damage scores in patients with SS when compared with control subjects.” (Page 2, Line 49-52) (highlighted in green)
2.“It was reported that “dry eye is a multifactorial disease of the ocular surface characterized by a loss of homeostasis of the tear film, and accompanied by ocular symptoms, in which tear film instability and hyperosmolarity, ocular surface inflammation and damage, and neurosensory abnormalities play etiological roles” by the international Dry Eye Workshop II (DEWS II) in 2017 [12].” (Page 2, Line 55-59) (highlighted in green)
3.“ The assessment of the histopathology to evaluate the ocular surface inflammation, damage, and neurosensory abnormalities which play etiological roles in dry eyes, has been problematic.”(Page 2, Line 59-60) (highlighted in green)
4.“The ability of in vivo confocal microscopy has been employed in investigation of the presence of inflammation and cytological changes in conjunction with ocular surface examinations such as vital staining, impression, and brush cytology [7-10,13-21].” (Page 2, Line 64-67) (highlighted in green)
5.“In vivo laser-scanning confocal microscopic observations of the conjunctiva in patients with SS revealed a decreased conjunctival epithelial cell density, increased conjunctival epithelial microcyst density, and increased conjunctival inflammatory cell density in a previous study [1].” (Page 2, Line 67-70) (highlighted in green)
- Tables 1 and 3 show various values for assessments made for control and SS patients; Tables 2 and 4 show values for assessments made for SS patients “before treatment” and after treatment with 3% diquafosal sodium for 3 months. The SS values listed for patients on Table 1 and 2 are different from those listed for SS patients before treatment on Tables 2 and 4. The authors should clarify this issue. The differences likely reflect different populations of patients.
(Reply 2)
Thank you for the nice comments and we apologize for the confusing presentation. As the reviewer mentioned, the populations of SS patients in Table 1 and 3 are different from those in Table 2 and 4. We combined Table 1 and 3, and also Table 2 and 4, in order to clarify these issues. Please confirm the modified Tables 1 and 2 in the revised manuscript. (Page 6-7) (highlighted in green)
- This reader kept referring back to controls while reading Tables 2 and 4. If possible, authors should consider combining Table 1 and 2 and Tables 2 and 4 to allow readers to see that while values for FL, RB, and BUT in Tables 1 and 2 and for inflammatory cells and sub-basal axon density for SS patients improve with treatment, most of these values remain significantly different from controls. One exception is superficial epithelial cell density and area which appear to be similar for SS patients after treatment compared to controls.
(Reply 3)
Thank you for the excellent comments. As above-mentioned, the populations of SS patients in Table 1 and 3 are different from those in Table 2 and 4. For that reason, we cannot combine Table 1 and 2, or Table 3 and 4. However, we can combine Table 1 and 3, or Table 2 and 4, in order to see easily for the readers. Please confirm the modified Tables 1 and 2 in the revised manuscript. (Page 6-7) (highlighted in green)
In addition, there were no significant differences in corneal superficial epithelial cell density (p>0.05) and area (p>0.05) between the patients with SS after the treatment and controls in this study.
- Another explanation for the positive correlation between dendritic cells and axon loss is the role that dendritic cells play in clearing axonal debris. As debris accumulates, dendritic cells activate to assist the corneal epithelial cells in removing it (see paper titled "Axonal Debris Accumulates in Corneal Epithelial Cells After Intraepithelial Corneal Nerves Are Damaged: A Focused Ion Beam Scanning Electron Microscopy (FIB-SEM) Study' in Exp Eye Res 2020 May; 194:107998). While immune cells may secrete cytokines that induce axon loss, they also may play roles in clearing debris after axon loss.
(Reply 4)
This is a very good comment and such essential information including the reference is now provided in the discussion section as follows;
“It was also reported that the dendritic (immune) cells mediated clearance of axonal debris from intraepithelial corneal basal nerves after injury [41]. In other words, while the dendritic (immune) cells may secrete the cytokines that induce the neural axon loss, they also may play roles in removal of axonal debris. This is another explanation for the positive correlation between the increased dendritic cells and decreased intraepithelial corneal basal nerves.” (Page 11, Line 344-348) (highlighted in green)
[41] Parlanti P, Pal-Ghosh S, Williams A, Tadvalkar G, Popratiloff A, Stepp MA. Axonal debris accumulates in corneal epithelial cells after intraepithelial corneal nerves are damaged: A focused Ion Beam Scanning Electron Microscopy (FIB-SEM) study. Exp Eye Res. 2020, 194, in press
- The authors use the classic canonical term for the sub-basal nerves. Consideration should be given to using the amended terminology proposed in the paper titled Corneal Epithelial “Neuromas”: A Case of Mistaken Identity? In the journal Cornea 2020;39:930–934. The "sub-basal nerves" and "sub-basal inflammatory cells" are actually located within the epithelium at the basal aspect of the corneal epithelial basal cells above the epithelial basement membrane and Bowman's layer. Using the classical terms can make it more difficult for colleagues to appreciate the anatomical relationships between the corneal epithelial cells, dendritic cells, basement membrane and Bowman's layer.
(Reply 5)
Thank you for your wonderful comments. We also agree with the reviewer that this information is important. In this study, we compare our confocal microscopic outcomes to those reported previously by Benitez-del-Castillo JM, Villani E, Tuominen ISJ, and Tuisku IS in dry eyes accompanied with Sjogren’s syndrome. However, all these studies used the same terminology “sub-basal nerves” and for purposes of comparison and ease of understanding, we preferred to keep this terminology which was also recognized in the 2017 International Dry Eye Workshop report for this imaging modality. Therefore, we also used the word of canonical “sub-basal nerves” in this study. However, we do recognize the importance of reviewer’s comment in relation to “intraepithelial corneal basal nerves” mentioned in the articles referenced under numbers 41 and we would like to refer to them in the revised paper.
According to the reviewer’s request, we added the below sentence in the discussion section as follows;
“In addition, Stepp MA et al. recommended the changes from the canonical “sub-basal nerves” to the proposed amended “intraepithelial corneal basal nerves” regarding the corneal nerve terminology in morphology [42].” (Page 11, Line 348-351) (highlighted in green)
[42] Stepp MA, Pal-Ghosh S, Downie LE, Zhang AC, Chinnery HR, Machet J, Girolamo ND. Corneal epithelial “Neuromas”: A case of mistaken identity? Cornea. 2020, 39, 930-934.
To the reviewer:
Thank you for your time and efforts during the revision of our manuscript.
We changed the order of manuscript and references.
Reviewer 2 Report
Congratulations to the authors. Really nice work.
Minor comments: could you modify the scale bars? It will be better without background. P3 L79 the word "criteria" was typed twice.
Author Response
Response to reviewer 2
Thank you for your time in reviewing our manuscript. We tried to revise the paper in accordance with the criticisms of the reviewer within our best providing point-by-point responses as follows:
Comments and Suggestions for Authors
Congratulations to the authors. Really nice work.
Minor comments: could you modify the scale bars? It will be better without background. P3 L79 the word "criteria" was typed twice.
(Reply)
We modified the scale bars in all figures (Figure 1, 2, and 3) according to the reviewer’ request. We also described the sentence of “Bar = 50 μm.” in the figure legends of manuscript (highlighted in blue). Please confirm them in the revised manuscript. (Page 7-8)
We deleted the typographical error “criteria” from the sentence (Page 3, Line 79) as follows.
“The diagnosis of dry eye was made by the following the Japanese consensus criteria: symptoms of dry eye, abnormality of tear production as determined by the Schirmer test (< 5 mm / 5 minutes), tear film instability as determined by the break-up time (BUT) (< 5 seconds), and positive ocular surface vital staining (with fluorescein and Rose Bengal dye) [23].” (Page 3, Line 94-97) (highlighted in blue)
To the reviewer:
Thank you for your time and efforts during the revision of our manuscript.
We changed the order of manuscript and references.